# First Evidence for a Role of Siglec-8 in Breast Cancer

**DOI:** 10.3390/ijms22042000

**Published:** 2021-02-18

**Authors:** Anna Trebo, Nina Ditsch, Tom Degenhardt, Christina Kuhn, Martina Rahmeh, Elisa Schmoeckel, Doris Mayr, Bastian Czogalla, Thomas Kolben, Sarah Meister, Sven Mahner, Udo Jeschke, Anna Hester

**Affiliations:** 1Department of Obstetrics and Gynecology, University Hospital, LMU Munich, 80337 Munich, Germany; Anna.trebo@campus.lmu.de (A.T.); nina.ditsch@uk-augsburg.de (N.D.); tom.degenhardt@med.uni-muenchen.de (T.D.); christina.kuhn@uk-augsburg.de (C.K.); martina.rahmeh@med.uni-muenchen.de (M.R.); bastian.czogalla@med.uni-muenchen.de (B.C.); thomas.kolben@med.uni-muenchen.de (T.K.); sarah.meister@med.uni-muenchen.de (S.M.); sven.mahner@med.uni-muenchen.de (S.M.); anna.hester@med.uni-muenchen.de (A.H.); 2Department of Obstetrics and Gynecology, University Hospital Augsburg, 86156 Augsburg, Germany; 3Institute of Pathology, LMU Munich, 80337 Munich, Germany; elisa.schmoeckel@med.uni-muenchen.de (E.S.); doris.mayr@med.uni-muenchen.de (D.M.)

**Keywords:** breast cancer, Siglec, Siglec-8, PPARγ, Gal-7, TA-MUC1, prognostic factor, targeted therapy

## Abstract

Sialic acid-binding immunoglobulin-like lectins (Siglecs) are involved in various immune cell-mediated diseases. Their role in cancer is poorly investigated, and research focusses on Siglec-expression on immune cells interacting with tumor cells. This study evaluates the role of Siglec-8 in breast cancer (BC). Siglec-8 expression was analyzed immunohistochemically on 235 primary BC cases and was correlated with clinical and pathological parameters and outcome. Cell culture experiments were performed with various BC cell lines. Siglec-8 was expressed in 215 BC cases and expression was lowest in triple-negative BC. It correlated with estrogen receptor-status, grading and the prognostic factors galectin (Gal)-7 and tumor-associated mucin-1 (TA-MUC1). However, Gal-7 and TA-MUC1 were only prognosticators for clinical outcome in the cohort expressing high (Immunoreactivity score IRS > 3) Siglec-8 levels but not in the low-expressing cohort. Siglec-8 knockdown led to a significantly reduced Gal-7 expression in MCF7 cells. All BC cell lines expressed low Siglec-8-levels, that could be elevated in MCF7 by Peroxisome proliferator-activated receptor (PPARγ)-stimulation. This study demonstrates that Siglec-8 is expressed in BC cells and correlates with known clinical and prognostic parameters. It is probably associated with Gal-7 and TA-MUC1 and might be regulated via PPARγ. Further analyses focusing on functional associations will clarify Siglec-8’s eligibility as a possible therapeutic target.

## 1. Introduction

Breast cancer (BC) is by far the most frequent malignant tumor in women. In 2018, about 2.1 million women were newly diagnosed with BC, with rising incidence since 1980 [1,2]. The World Health Organization (WHO) estimates malignancies to become the leading causes of mortality after 2030 and BC to become the second leading cause of tumor-related deaths [2]. 

BC can be classified into different intrinsic subtypes [3]. Daily clinical practice typically uses a surrogate classification of the five subtypes on the basis of histological and molecular characteristics [4]: (1) Luminal A-like tumors show a strong expression of estrogen (ER) and progesterone (PR) receptors, low proliferation rates and a good prognosis. (2) Luminal B-like human epidermal growth factor receptor 2 (HER2)-negative tumors show lower hormone receptor (HR) expression and higher grading than Luminal A-like tumors and have an intermediate prognosis. Tumors that show an amplification of HER2 can be further classified based on HR expression (3) in Luminal B-like HER2-positive and (4) in HER2-positive non-luminal tumors. (5) Triple-negative BC (TNBC) lacks the expression of both HR and HER2 and show the worst prognosis of all biological BC subtypes [3,5]. Even though therapy has improved over the past years [1], new therapeutic strategies aiming at specific targets are still needed [6,7,8]. 

Sialic acid-binding immunoglobulin-like lectins (short Siglecs) form a group of receptors within the subfamily of I-type (immunoglobulin-type) lectins. I-type lectins, first described in 1995 [9] as integral membrane proteins, often occur with large cytosolic domains and established phosphorylation sites (like Cluster of Differentation molecule 33, 22 (CD33 & CD22) and Myelin Associated Glycoprotein (MAG)). The subfamily of those having the N-terminus consisting of a sialic acid binding lectin domain and whose C-terminal cytoplasmic region typically, but not uniformly, contains conserved signaling domains, was suggested to be called Siglecs in 1998 [10]. Siglecs can be divided into two subgroups: the first is an evolutionary conserved group, consisting of Siglec-1, -2 (CD22), -4 and -15. The second group is formed by Siglec-3 and further CD33 (Siglec-3)-related Siglecs (like Siglec-5 and Siglec-8). These are mainly expressed in various cells of innate immunity (granulocytes, monocytes and macrophages) [11,12]. The various roles of Siglecs in ligand recognition and binding involving cell–cell interactions, but also in intracellular signaling and immune system regulation [12], suggest that they have a major impact on disease pathophysiology, which makes them useful as biomarkers or potential targets. Recently, the main focus of attention when investigating Siglecs was set on eosinophil and mast cell Siglecs [13,14] and on their role in the therapy of diverse pathologies, like eosinophilic gastrointestinal disorders [15]. Some antibodies targeting Siglecs (like Siglec-3 and -2) are already approved or in clinical trials for treatments of immune cell-mediated diseases [16]. Siglecs were also studied in relation to the current COVID-19 pandemic: a large investigation on the glycan structures of the receptor binding domain of the SARS-CoV2 spike glycoprotein was performed. The different possible interacting epitopes have been analyzed and characterized and interactions of the glycans with different human lectins like galectins (galectins-3, -7 and -8) and Siglecs (Siglec-8, Siglec-10) have been evaluated [17].

However, only very little is known about the role of Siglecs in general in the development, growth or repression of tumors. First research about Siglecs and tumorigenesis focused on tumor-associated macrophages (TAMs) and the influence of Siglecs in their interaction with tumor cells. It was discovered that Siglec-1 is involved in an auto-regulatory loop between TAMs and cancer cells in aggressive BCs, identifying Siglec-1 and C-C Motif Chemokine Ligand 8 (CCL8) expression together as prognostic markers for poor survival [18]. Furthermore, it was recently shown that CD169+ (Siglec-1) macrophages in the tumor microenvironment promote the progression of TNBC, at least partially, by inhibiting the antitumor immunity of T cells against BC cells [19].

Regarding specifically Siglec-8, it was initially found only on eosinophils, appearing to be the first eosinophil-specific transmembrane receptor [20]. It is now known that Siglec-8 is expressed by eosinophils, mast cells and, in small amounts, by basophils [14]. It is upregulated in the chronically inflamed airway, where it can inhibit inflammation when binding to ligands [21,22]. Recently, many studies have shown the influence of Siglec-8 in eosinophilic disorders [23], especially as a biomarker of eosinophil involvement in allergic and eosinophilic diseases [24]. But Siglec-8 was also detected as a late maturation marker on eosinophils and basophils in patients with chronic eosinophilic leukemia, chronic myelogenous leukemia and on malignant and non-malignant bone marrow mast cells [25]. In eosinophils, interleukin-5 can upregulate Siglec-8 surface expression [26], and Siglec-8 crosslinking with specific antibodies induces eosinophil cell death [27,28]. Interleukin-5 (IL-5) priming enhances the Siglec-8-mediated apoptosis in eosinophils. In activated eosinophils, ligation of Siglec-8 leads to a reactive oxygen species-dependent enhancement of IL-5-induced Mitogen-Activated Protein Kinase (ERK) (ERK) phosphorylation, which results in a biochemically regulated eosinophil cell death [29]. Not only interleukin-5 but also interleukin-33 is effective in enhancing Siglec-8-mediated eosinophil apoptosis and can synergize with interleukin-5 [30]. These pathways play an important role in hypereosinophilic disorders like asthma [31]. 

It’s important role makes Siglec-8 suitable as a target for treatment of eosinophil- and mast cell-related diseases, such as asthma, chronic rhinosinusitis, chronic urticaria, hypereosinophilic syndromes, mast cell and eosinophil malignancies and eosinophilic gastrointestinal disorders [32]. The attention on Siglec-8 as a potential target in many diseases led to the development of a ligand targeting liposomes to cells expressing Siglec-8 [33] and to the establishment of anti-Siglec-8 antibodies. It was shown that anti-Siglec-8 antibodies, in the presence of secondary antibodies, induce apoptosis of eosinophils [16]. Siglec-8 antibodies are currently investigated in mast cell and eosinophilic disorders [34]. In this context, it was also seen that intravenous immunoglobulins contain naturally occurring antibodies against Siglecs. They might be necessary as an immunoregulatory mechanism [35]. 

Only little is known about Siglec-8 and its role in tumor biology so far. Limited data is available in clear cell renal carcinoma and in gastric cancer [36,37]. 

The aim of the study was to create first evidence of a possible role of Siglec-8 in BC.

## 2. Results

### 2.1. Siglec-8 Expression in Breast Cancer and Correlation to Different Clinical and Pathological Characteristics

#### 2.1.1. Siglec-8 Expression in BC 

Expression of Siglec-8 could be evaluated in 226/235 tissue sections (could not be evaluated in 9 sections due to technical issues). Of these cases, 11 showed no Siglec-8 expression, and the median immune reactivity score (IRS) of Siglec-8 expression was 6. The distribution of the IRS is shown in Table 1. Nuclear staining could not be observed. 

#### 2.1.2. Correlation of Siglec-8 Expression with Clinical Characteristics, Histopathological BC Subtypes and Grading

The extent of Siglec-8 expression (measured as IRS) did not correlate neither with the clinical parameters, tumor size and lymph node status, nor with patient age (see Spearman correlation analyses in Appendix A). 

The extent of Siglec-8 expression (IRS) correlated significantly with the histopathological subtype (correlation coefficient (CC) −0.18, *p* = 0009). Kruskal–Wallis test and boxplots analysis showed that Siglec-8 expression was significantly higher in tumors of no special type (NST) compared to non-NST tumors (*p* = 0.009) (Figure 1a). 

There was a correlation of borderline significance (CC = 0.152, *p* = 0.059) between Siglec-8 expression and tumor grading (G). Kruskal–Wallis test showed that Siglec-8 expression was higher in higher tumor grading (G1: median Siglec-8 IRS: 3, G2/3 median Siglec-8 IRS: 6, *p* = 0.007, Figure 1b). Exemplary immunohistochemical Siglec-8-stainings in tumors with different gradings are shown in Figure 2. Information about tumor grading is only available in about 70% of all patients, as certain histological subtypes (e.g., lobular, medullar) were not routinely graded at the time these patients were diagnosed with BC.

#### 2.1.3. Correlation of Siglec-8 Expression with the Biological BC Subtype and Further Prognostic Factors

Spearman analysis revealed that Siglec-8 expression did correlate to ER status (CC = 0.147, *p* = 0.027) but not to PR status, HER2 amplification or the biological subtype (Spearman analysis in Appendix A). In the Kruskal–Wallis analysis, the Siglec-8 expression was significantly higher in ER-positive compared to ER-negative tumors (ER-positive: median Siglec-8 IRS 6 vs. in ER-negative: median Siglec-8 IRS 4, *p* = 0.027, Figure 3a), but was not significantly different concerning PR status (*p* = 0.098, data not shown) or HER2 status (*p* = 0.103, data not shown). The Siglec-8 expression did not differ significantly comparing the different biological subtypes among each other, with partly small sample sizes of single subtypes (Kruskal–Wallis analysis, *p* = 0.103, Figure 3c). However, comparing TNBC to all other subtypes, Siglec-8 expression was significantly lower in TNBC in comparison to all other subtypes (TNBC: median Siglec-8 IRS 4 vs. in the other subtypes: median Siglec-8 IRS 6, *p* = 0.040, Figure 3b). 

Siglec-8 expression was also compared to the expression of the tumor-associated epitope of mucin-1 (TA-MUC1, measured by Gatipotuzumab-staining) and to the cytoplasmic levels of Galectin-7 (Gal-7)—both prognostic factors that have already been evaluated in this cohort by our group before [38,39]. Siglec-8 expression correlated significantly with TA-MUC1 expression in the cytoplasm (CC = 0.14, *p* = 0.039, see Appendix A, no correlation was found of Siglec-8 with membranous TA-MUC1 expression CC = 0.017, *p* = 0.803) and to Gal-7 expression (CC = 0.298, *p* < 0.001, see Appendix A).

### 2.2. Correlation of Siglec-8 Expression with Survival in BC Patients

Median overall survival (OS), progressive-free survival (PFS) and distant disease-free survival (DDFS) was not reached (NR) in the whole cohort. For survival analyses, tumors were categorized in “Siglec-8 high” and “Siglec-8 low” expressing tumors using receiver operating characteristic (ROC)-curve analysis. An IRS of >3 was considered as high expression of Siglec-8 and IRS of 0–3 as low. 

#### 2.2.1. Survival Analysis Concerning Siglec-8 Expression

In the overall cohort, Siglec-8 expression (IRS > 3 vs. IRS 0–3) did not correlate with differences in survival regarding PFS (*p* = 0.971), DDFS (*p* = 0.941) or OS (*p* = 0.850) (Appendix A).

In subgroup survival analysis, there was an association of borderline significance between a high Siglec-8 expression and an impaired PFS and DDFS in the node-positive subgroup (PFS: *p* = 0.237, Appendix A, DDFS: *p* = 0.117, Appendix A). On the contrary, in the pN0 subgroup, PFS seemed to be superior in tumors with high Siglec-8 expression (*p* = 0.061, Appendix A). In any other subgroups (like ER-, PR-positive/negative or different grading), no association of Siglec-8 with PFS or DDFS could be revealed. Regarding OS, a tendentially impaired OS in Siglec-8 high-expressing tumors was observed in HER2-positive patients (*p* = 0.118, Appendix A, but not in HER2-negative, data not shown) and in ER-negative patients (*p* = 0.153, Appendix A, but not in ER-positive, data not shown).

Due to the correlations of Siglec-8 with Gal-7 and TA-MUC1, survival analyses were also performed regarding these parameters in the context of Siglec-8 expression.

#### 2.2.2. Survival Analysis Using Combined Siglec-8 and Gal-7 Expression

Earlier data described that cytoplasmic Gal-7 expression is a prognostic factor for an impaired PFS and DDFS [39]. An IRS > 6 for Gal-7 expression was considered as Gal-7-positive, and an IRS of 0–6 for Gal-7 as negative.

When Siglec-8 expression was included in the Gal-7 survival analysis, it was revealed that the prognostic relevance of Gal-7 was only present in the Siglec-8 high-expressing subgroup. In patients with high Siglec-8 expression, high Gal-7 expression was significantly associated with an impaired PFS (*p* = 0.023, Figure 4a). In Siglec-8 low-expressing patients, PFS did not differ significantly between Gal-7 high- and low-expressing patients (*p* = 0.276, Appendix A). So, patients with both a high Siglec-8 and a high Gal-7 expression showed a significantly impaired PFS compared to all other combinations of high/low either Siglec-8 or Gal-7 expression (median PFS in Gal-7 high and Siglec-8 high 9.76 years, in the others NR, *p* = 0.032, Figure 4c). This subgroup constituted 15.7% of all patients. 

Similar effects could be demonstrated regarding DDFS, where Gal-7 had an association of borderline significance with DDFS only in the Siglec-8 high-expressing subgroup (*p* = 0.059, Figure 4b, not in Siglec-8 low-expressing patients, Appendix A). Patients with both a high Siglec-8 and a high Gal-7 expression also showed a significantly impaired DDFS compared to all other subgroups (median DDFS in all subgroups NR, *p* = 0.039, Figure 4d).

Regarding OS, no differences could be observed for Gal-7, neither in the overall cohort (as already previously described in [39]) nor in the subgroups of high and low Siglec-8 expression (Appendix A). 

#### 2.2.3. Survival Analysis Using Combined Siglec-8 and TA-MUC1 Expression

As described before, membranous TA-MUC1-expression measured by Gatipotuzumab-staining is a prognostic factor for an improved OS, while cytoplasmic TA-MUC1-expression is not [38]. In the current analysis, in addition to the published data, an association of borderline significance of membranous TA-MUC1 expression with an improved DDFS (*p* = 0.066, Figure 5a) could be shown. According to previously published data, an IRS > 2 for membranous TA-MUC1 expression was considered as TA-MUC1-positive, and an IRS of 0–2 for membranous TA-MUC1 as negative.

When Siglec-8 expression was included in the TA-MUC1 survival analysis, it could be demonstrated that the prognostic relevance of membranous TA-MUC1 regarding OS was only present in the Siglec-8 high-expressing subgroup: in patients showing a high Siglec-8 expression, a high expression of membranous TA-MUC1 was significantly associated with an improved OS (*p* = 0.017, Figure 5c). In Siglec-8 low-expressing patients however, membranous TA-MUC1 was not significantly associated with OS (*p* = 0.443, Appendix A). 

Siglec-8 furthermore improved the prognostic accuracy of membranous TA-MUC1 regarding DDFS; however, contrary to OS data in the Siglec-8 low-expressing subgroup, in the Siglec-8 low-expressing subgroup, high membranous TA-MUC1 expression was significantly associated with an improved DDFS (*p* = 0.039, Figure 5b). In Siglec-8 high-expressing patients, DDFS did not differ significantly between membranous TA-MUC1-positive and -negative patients (*p* = 0.307, Appendix A). 

Regarding PFS, membranous TA-MUC1 was not a prognostic factor neither in the overall (*p* = 0.102) nor in the Siglec-8 low- (*p* = 0.132) or high-expressing cohort (*p* = 0.205, Appendix A). As previously described for the overall cohort [36], cytoplasmic TA-MUC1 expression was also not a prognostic factor for survival in Siglec-8 low- or high-expressing patients (data not shown).

### 2.3. In vitro Experiments with BC Cell Lines

#### 2.3.1. Siglec-8 Expression in Different BC Cell Lines on mRNA and Protein Level 

The role of Siglec-8 in BC was further investigated using cell culture models. The expression of Siglec-8 on mRNA level was low in all cell lines investigated (mean relative expression 2^-ΔΔCT^ 0.9 in MDA-MB 231, 1.4 in T-47D, 0.8 in MCF7, *n* = 3, Figure 6a). On the protein level (*n* = 3, Figure 6b), a weak Siglec-8 expression could be detected in MDA-MB231 cells (45 % normalized to β-Actin as a loading control), in MCF7 (24 %) and T47D (20 %) cells. Exemplary Western Blots are shown in Figure 6b.

#### 2.3.2. Siglec-8 Knockdown and Gal-7 Expression

Due to the correlation of Siglec-8 and Gal-7 expression and their combined prognostic association, a possible influence of Siglec-8 on Gal-7 expression was investigated. When silencing Siglec-8 with three different small interfering RNAs (siRNAs), a significantly downregulated Gal-7 expression in MCF7 cells to 94% (siRNA A, *p* < 0.001, *n* = 3), 90% (siRNA B, *p* < 0.01, *n* =3) and 86% (siRNA C, *p* = 0.231, *n* = 3) compared to the Siglec-8 wildtype control cells was observed (Figure 7). 

#### 2.3.3. Siglec-8 mRNA Expression after Stimulation with β-Estradiol and Rosiglitazone

As ER status and Siglec-8 expression correlated in the Immunohistochemistry (IHC) analysis, it was examined whether stimulating the ER influences Siglec-8 expression. Stimulating the cells with β-estradiol for 24 or 48 h did not result in any differences in the Siglec-8 expression (data not shown).

Furthermore, literature research revealed a Peroxisome proliferator-activated receptor (PPAR)γ-binding site in the Siglec-8 gene in GeneCards [40]. Therefore, the influence of the PPARγ agonist rosiglitazone on Siglec-8 expression on mRNA level was analyzed. Stimulation of MCF7 cells with 1 µg/ml rosiglitazone did not influence Siglec-8 expression. Stimulation with 10 µg/mL rosiglitazone raised the relative Siglec-8 expression up to 171% (*p* = 0.026, *n* = 3) after 1 h and up to 189% (*p* < 0.001, *n* = 3) after 2 h compared to the unstimulated control (normalized to GAPDH as housekeeper gene, Figure 8). 

## 3. Discussion

The receptor family of Siglecs is mainly known for its role in immune-related diseases. Siglec-8 has been under investigation as a therapeutic target in eosinophilic diseases [16,32]. Regarding tumorigenesis, Siglecs have been recently shown to be expressed on TAMs and might contribute to the tumor cell–macrophage interaction. However, only very little data is available about the role of Siglecs expressed on tumor cells. In this study, we aimed to create first evidence of a possible role of Siglec-8-expression in BC.

We observed that Siglec-8 is expressed in varying intensity in BC cells without nuclear staining. This expression pattern highlights Siglec-8’s function as a transmembrane protein [20]. A comparable staining pattern was observed in eosinophils [41]. Interestingly, only two studies used IHC to determine Siglec-8 expression in cancers (in renal and gastric cancer [36,37]), with both showing similar expression patterns as in our BC panel.

We observed higher Siglec-8 expression levels in BCs with a higher grading, which indicates that Siglec-8 expression might be induced when de-differentiation of tumor cells occurs. Siglec-8 expression was significantly higher in ER-positive than in ER-negative tumors and was lowest in TNBC. 

Siglec-8 positivity or negativity was not associated with survival rates. However, Siglec-8 expression correlated to the previously identified prognostic factors cytoplasmic Gal-7 levels (negative prognostic factor [39]) and TA-MUC1 expression (membranous expression: positive prognostic factor, cytoplasmatic expression: no prognostic association [38]) in BC.

Evaluating these prognostic factors in the context of Siglec-8 expression, we could demonstrate that a high Gal-7 expression was associated with an impaired PFS in the Siglec-8 high-expressing subgroup. In the Siglec-8 low-expressing subgroup, Gal-7 occurred, but not as a prognostic factor. Furthermore, Siglec-8 knockdown led to a reduced Gal-7 expression, which indicates an interaction of these two proteins. In GeneCards, an interaction of Gal-3—which belongs to the same group as Gal-7—and Siglec-8 is described [40]. The interaction of Gal-7 and Siglec-8 might be involved in the mechanism of how Gal-7 levels in tumor cells are regulated: they can be increased by either the induction of mRNA expression or an extracellular to intracellular transfer of Gal-7 [42]. In general, it is known that N-acetyllactosamins (LacNAc) epitopes bind to galectins like Gal-1, Gal­-3 and Gal-7 [43]. For Gal-1, an important role of LacNAcs in the extracellular to intracellular transfer has been shown: extracellular glycans that bear LacNAc epitopes bind Gal-1 and trap it extracellularly. An α-2,6-sialylation of these LacNAc epitopes inhibits the Gal-1 binding and drives the intracellular and then nuclear transfer of Gal-1 [44]. A similar mechanism probably exists for Gal-7. Siglec-8 is known to bind sialylated LacNAcs [45]. By doing so, it might stabilize the sialylated extracellular LacNAc epitope and promote the liberation of extracellularly bound Gal-7, which could then be transferred intracellularly. The intracellular level of Gal-7 itself can regulate Gal-7 mRNA expression [42]. This could be an explanation as to how Siglec-8 knockdown leads to a reduced mRNA expression of Gal-7; however, further functional analyses will have to follow to thoroughly analyze these suggested pathways. 

On the other hand, the positive prognostic association of membranous TA-MUC1 with OS was also only present in the Siglec-8 high-expressing subgroup. In the Siglec-8 low-expressing subgroup, no associations of TA-MUC1 and OS could be seen. The association between TA-MUC1 and Siglec-8 seems less consistent, as contrary to OS-data, an association of membranous TA-MUC1 expression with DDFS was only present in the Siglec-8 low-expressing subgroup. No data is currently available in the literature about an interaction between Siglec-8 and TA-MUC1. However, when MUC1 is expressed on tumors, it is frequently sialylated [35], and Siglecs are known to bind sialic acid structures. A binding of Siglec-9 on macrophages to MUC1 on tumors has been described [46]. It might be that Siglec-8 on BC cells co-locates with and therefore “presents” TA-MUC1 or that Siglec-8 binds TA-MUC1 and “transports” it from the cytoplasm to the membrane. Siglec-8 expression correlated to the cytoplasmic TA-MUC1 levels in our study. TA-MUC1 in the cytoplasm is associated with an impaired survival when directly compared to membranous TA-MUC1 [38]—Siglec-8 might be involved in a shuttling of TA-MUC1 from cytoplasm to membrane. These hypotheses about an interaction between TA-MUC1 and Siglec-8 could explain why TA-MUC1 does not show a prognostic association regarding OS in the Siglec-8-negative subgroup. A further mechanism could include galectins, as they—including Gal-7—were also found to bind MUC1 [47]. However, how the contradictory prognostic effects of cytoplasmatic Gal-7 and membranous TA-MUC1 might be mediated by Siglec-8 needs further research. 

To summarize, survival analyses from our study suggest an association of Siglec-8 with both positive and negative prognostic factors in BC, and a high Siglec-8 expression was especially present in Luminal-like breast cancer. So, inhibiting Siglec-8 in addition to endocrine therapies might be a therapeutic strategy after functional associations have been further clarified. Here, cell culture models can help to study functional effects of Siglec-8. 

Although we found a strong Siglec-8 expression in IHC of BC tumors, the Siglec-8 expression on mRNA and protein levels in the BC cells lines we analyzed was quite low. Stimulation with estradiol did not influence Siglec-8 expression. After finding PPARγ binding sites in the transcription factor in the Siglec-8 gene promoter in GeneCards [40], we also stimulated BC cell lines with a PPARγ agonist. This led to a stable mRNA elevation of Siglec-8. Siglec-F in mice is assumed to be the equivalent to Siglec-8 in humans. Therefore, experiments in vivo mouse models could be done to verify the effect, even though there are some differences in expression patterns [48].

Peroxisome proliferator-activated receptor-γ (PPARγ) is a ligand-activated nuclear hormone receptor that functions as transcription factor and is over-expressed in many tumor types, including BC [49,50]. Effects of PPARγ ligands in BC have not been fully understood yet, but data suggest that ligands like Rosiglitazone inhibit proliferation and induce apoptosis [51]. Rosiglitazone was investigated in clinical trials [52] without achieving breakthroughs, yet [53]. The effectiveness of PPARγ therapy could be improved by better understanding the proteins involved in related pathways such as Siglec-8. The numerous effects of anti-PPARγ therapy might include the mechanism by which PPARγ-antagonism reduces Siglec-8 expression.

The role of Siglecs in tumors, including possible therapeutic targeting, is currently being investigated. This research focuses on the role of Siglecs as targetable immune checkpoints [54]. The CD33 group of Siglecs was found to play a major role as immune checkpoint molecules in the tumor-microenvironment of BC: inhibiting Siglec-7 (expressed on eosinophils and Natural killer (NK) cells) could induce NK cell lysis of tumor cells [55]. Siglec-9 was expressed on TAMs and has been shown to interact with MUC1 on tumor cells [46,56]. Furthermore, TAMs express high levels of Siglec-10, which interacts with CD24. CD24 can be the dominant innate immune checkpoint in ovarian cancer and BC and is a promising target for cancer immunotherapy [57]. Siglec-15, identified on antigen-presenting cells as an inhibitor of T cell activation, is targeted with an antibody in an early clinical trial for advanced solid tumors. Targeting Siglec-15 was tested in a system linked to the HER2-targeting antibody and an NK cell-mediated tumor cell killing was tested in vitro and in vivo [58].

However, all these studies aim to target Siglecs on immune cells in the tumor microenvironment. In contrast, our study focused on the role of Siglec-8 on the tumor itself. Regarding the role of Siglec-8 on tumors, Siglec-8 expression (measured by IHC) was identified as a potential independent prognostic biomarker of clear cell renal cell carcinoma, where a high expression correlated with an impaired OS and DDFS [36]. In contrast, low Siglec-8 expression (IHC) was an independent poor prognosticator for OS in patients with gastric cancer after surgical resection. This was especially seen in higher TNM stages, and the authors suggested that low Siglec-8 expression could be used as a marker to identify patients needing more aggressive adjuvant therapies [37].

Interestingly, already, in 2000, an anti-CD33 (= Siglec-3) antibody as part of an antibody–drug conjugate was approved for treating acute myeloid leukemia [59]. After the role of Siglec-8 has been fully clarified in BC, using it in an antibody–drug conjugate could be an option.

Our study gives first evidence about a role of Siglec-8 expression in BC. Further studies will have to clarify functional aspects to evaluate its role as a possible therapeutic target. 

## 4. Materials and Methods 

### 4.1. Patients

For this study, 235 formalin-fixed paraffin-embedded (FFPE) primary BC samples were examined (patient characteristics are shown in Table 2). All patients were diagnosed with primary non-metastatic BC (M0) and underwent surgery at the Department of Gynecology and Obstetrics, Ludwig-Maximilians-University Munich, Germany, from 1998 until 2000. Women with benign tumors of the breast were excluded from the study. Mean patients’ age at the time of surgery was 58.2 ± 13.3 years. 

Histopathological subtype (NST vs. non-NST), tumor grading (G1-3) according to the Elston and Ellis criteria (1993) [5,60,61] and staging using the TNM-System [62] (T for tumor size, N for the lymph node status and M for metastasis), were gathered by a gynecological pathologist. At the time of primary diagnosis of BC of these patients, certain histological subtypes (e.g., lobular, medullar) were not routinely graded. Therefore, information about tumor grading is only available in about 70 % of all patients, so the results have to be regarded with limited reliability.

Clinical and follow-up data, survival data, lymph node status, presence of metastases, ER/PR results and HER2 detection were retrieved from patients’ charts and from the Munich Cancer Registry. HER2 positivity is clearly defined by the DAKO (Agilent Technologies, Waldbronn, Germany) Scoring system (DAKO, HER2 Fluorescence in situ hybridization (FISH) pharmDx™ Kit). As HER2 status was not determined routinely in Germany before 2001, it was retrospectively assessed for patients who had surgery before 2001. HER2 status was determined as recommended in the national guidelines, i.e., by DAKO Score and FISH analysis in cases of DAKO 2+. 

Endpoints regarding the survival data were defined as follows: OS = overall survival, period of time from the date of surgery until the date of death or date of last follow-up, PFS = progression free survival, period of time until local recurrence or metastasis were diagnosed and DDFS = distant disease-free survival: period of time until metastasis is diagnosed.

### 4.2. Immunohistochemistry

Paraffin-embedded BC tissue samples were analyzed by immunohistochemistry. The samples were fixed in neutral buffered formalin and embedded in paraffin after surgery. For histopathological investigations, tissue sections (3 µm) were deparaffinized in Roticlear (Carl Roth GmbH + Co. KG) for 20 min and then the endogenous peroxidase was inactivated with 3% hydrogen peroxide (VWR International GmbH) in methanol. The slides were rehydrated in a descending gradient of ethanol (100%, 75% and 50%) and prepared for epitope retrieval in a pressure cooker for 5 min in sodium citrate buffer (0.1 mol/L citric acid, 0.1 mol/L sodium citrate, pH 6.0). After washing in distilled water and phosphate-buffered saline (PBS), all tissue slides were blocked using a blocking solution (Reagent 1; ZytoChem Plus HRP Polymer System (Mouse/Rabbit); Zytomed Systems GmbH, Berlin, Germany) for 5 min at room temperature (RT) in order to block non-specific binding of the primary antibodies. Then, slides were incubated with Siglec-8 primary antibody (rabbit, polyclonal; Novusbio, NBP1-31141) diluted in PBS Dulbecco (Biochrom GmbH, 1:250) for 16 h at 4 °C. Afterwards, the staining specimens were incubated in post-block reagent (Reagent 2) and ZytoChem Plus HRP Polymer System, Mouse/Rabbit (Reagent 3), according to the manufacturer’s protocol. All slides were washed in PBS after every incubation step. The slides were then stained with 3,3′-diaminobenzidine chromogen (DAB; Dako, Glostrup, Denmark) for visualization and counterstained in Mayer acidic hemalun. After dehydrating in an ascending ethanol gradient and Roticlear, they were cover-slipped with ROTI^®^ Mount (Carl Roth GmbH + Co. KG). Appropriate tissue slides were used as positive controls (colorectal cancer). To obtain expression results, the semiquantitative immunoreactive score (IRS, Remmele and Stegner 1987 [63]) was performed using a Leitz Diaplan microscope (Leitz, Wetzlar, Germany). The score was optically obtained by multiplying the predominant staining intensity (0: none, 1: low, 2: moderate, 3: strong) and the percentage of positively stained cells (0 = 0%, 1= 1−10%, 2 = 11−50%, 3 = 51−80% and 4 = 81−100% stained cells). Images were taken with a CCD color camera (JVC, Victor Company of Japan, Yokohama Japan).

### 4.3. Cell Culture and Drugs

The BC cell lines used in this study were obtained from the American Type Culture Collection (Manassas, VA, USA).

The following BC cell lines were used in this study: MCF7 (adenocarcinoma cells, ER/PR-positive, HER2-negative), MDA-MB-231 (adenocarcinoma cells, ER/PR-negative, HER2-negative) and T-47D (ductal carcinoma cells, ER/PR-positive, HER2-negative).

The cells were maintained in RPMI 1640 Medium + GlutaMAX (Thermo Fisher, Carlsbad, CA, USA) complemented with 10% FBS (Thermo Fisher). The cells were incubated at 37 °C and 5% CO2 saturation. As preparation for each experiment, the cells were counted using Neubauer cell chambers, seeded in 6–96-well plates and incubated overnight. After 24 h, the cell culture medium was replaced by Reduced-Serum Medium (Gibco, Opti-MEM) with either β-Estradiol (water-soluble E4389 Sigma Aldrich, Darmstadt, Germany) (1000 µMol) for 24 or 48 h or Rosiglitazone (R2048 Sigma-Aldrich) in concentrations of 1, 10, 20, 50 or 100 µg/mL for 1 or 2 h. Knockdown of Siglec-8 was performed by using Lipofectamine RNAiMAX Transfection Reagent (13778100, Invitrogen, Karlsruhe, Germany) according to the manufacturer’s instructions with 5 or 10 nM siRNA Siglec-8 (OriGene, Herford, Germany, SR309023). Three different siRNAs were provided: A, B and C, and one scrambled negative control siRNA, and they were incubated for 48 h.

### 4.4. mRNA Expression

The Siglec-8 expression on mRNA level in MCF-7, MDA-MB-231 and T-47D BC cell lines was determined using quantitative real-time (RT)- Polymerase chain reaction (PCR). The RNeasy Mini Kit (Qiagen, Hilden, Germany) was used to obtain the total RNA from cultured cells. The RNA was converted to cDNA with a cDNA Synthesis Kit (Biozym 331470L) according to the manufacturer’s instructions. For the RT-PCR, a 20 µL reaction mixture was made up as follows: 1 µL TaqMan^®^ Gene Expression Assay 20× (Applied Biosystems, Darmstadt, Germany, target GAPDH Nr Hs99999905_m1, target Siglec-8 Nr. Hs00274289_m1, target Gal-7 Nr. Hs00170104_m1, primer sequences are not available due to the use of a commercial assay), 10 µL TaqMan^®^ Fast Universal PCR Master Mix 2× (Applied Biosystems, Darmstadt, Germany), 1 µL cDNA template and 8 µL RNase free water per sample. RT-PCR was performed on a 96-well plate (Applied Biosystems) covered with an optical adhesive film. A 7500 Fast Real-Time PCR system (Applied Biosystems) was used to run the PCRs. Initially, for the enzyme activation, heating to 95 °C for 20 s was performed, followed by 40 qPCR-cycles of 3 s of denaturation at 95 °C and annealing for 30 s at 60 °C. 

### 4.5. Western Blot

Western Blot analyses were performed to analyze the protein expression level of Siglec-8. Different unstimulated cell lines (MDAMB231, MCF7 and T47D) were analyzed. For protein extraction, RIPA buffer (Sigma-Aldrich: R0278 with Protease Inhibitor P8340) was added and the samples were kept on ice for 30 min to obtain cell lysates. After centrifugation, supernatant with proteins were prepared in 4× Laemmli loading buffer. The samples and a protein marker (VWR peqGOLD, Darmstadt, Germany) were then loaded on a 10% polyacrylamide gel (SDS-PAGE) and separated at a constant voltage of 70 V for 2 h. Afterward, the proteins were transferred to a polyvinylidene fluoride membrane (Bio-Rad, Redmont, WA, USA) for 75 min at 145 mV and 4 °C. The membrane was blocked in 5% milk powder (diluted in sodium Tris Buffered Saline (TBS)/Tween) for 1 h and then incubated with the primary antibody overnight at room temperature. The primary antibody concentrations were used as in the following: mouse monoclonal anti β-actin antibody (1:1000; Sigma-Aldrich, Darmstadt, Germany, A5441), antiSiglec-8 antibody (1:1000, Novusbio NBP1-31141). After washing the membranes three times for 10 min in TBS/Tween, the samples were incubated with the secondary antibodies (concentration: 1.5 µg/mL, AntiMouse BA2000 and AntiRabbit BA1000, RT-biotinylized) for 45 min at room temperature. After another washing step (3 × 10 min), an incubation of 20 min with Vectastain ABC AmP Reagent (AK-6000, Vector Laboratories) followed by another washing step (3 × 10 min) was performed. The membrane was then incubated in Tris(hydroxymethyl)aminomethane (TRIS)-buffer for 5 min and bands were dyed using the color development BCIP/NBT Substrate (SK-5400, Vector Laboratories) in 0.1 M Tris for 10–30 min. Reaction was stopped in aqua dest and membrane was dried and protected from light. 

### 4.6. Statistical Analysis

Data analyses were performed with SPSS Statistics 25 (Armonk, NY: IBM Corp.). *p*-values lower than 0.05 were considered as statistically significant. Correlations between staining results and ordinal variables were tested with Spearman’s rank correlation coefficient. Group comparisons regarding IRS of galectins between different clinical and pathological subgroups were tested with the Kruskal–Wallis test and displayed as boxplot graphs. Survival times between different groups were compared by Kaplan–Meier analysis, and differences were tested for significance by Log-rank (Mantel–Cox) tests. Cox regression analysis was used to determine independency of prognostic factors.

Concerning survival analysis dependent on Siglec-8 expression, patients were grouped in high and low expression. Cut-off points were selected considering the distribution pattern of IR scores in the collective. Therefore, the ROC curve was drawn which is considered as one of the most reliable methods for cut-off point selection. In this context, the ROC curve is a plot representing sensitivity on the y-axis and 1-specificity on the x-axis. Consecutively, Youden index, defined as the maximum (sensitivity + specificity − 1), was used to find the optimal cut-off, maximizing the sum of sensitivity and specificity. The cytoplasmatic Siglec-8 expression was regarded as low with an IRS 0–3 and as high with IRS > 3. 

Real-Time PCR results were analyzed with Microsoft Excel by using the comparative 2^−ΔΔCT^ method in order to obtain mRNA expressions and T-Test was performed to calculate significance. GAPDH was used as endogenous controls for ΔCT-values and the results are means of triplicates. 

### 4.7. Ethics Statement

All tissue samples used for this study were left-over material from the archives of the Department of Gynecology and Obstetrics, Ludwig-Maximilians-University (LMU), that had initially been collected for histopathological diagnostics. All diagnostic procedures had already been completed when histopathological investigations for the current study were performed. Patients’ data have been anonymized and the author was blinded to the patients’ information during the analysis. The study was approved by the Ethics Committee of LMU Munich (reference number: 048-08; 2006). All experiments were performed according to the standards set in the declaration of Helsinki, 1975 [64].

## 5. Conclusions

Our data provide first evidence for a role of Siglec-8 in BC. Siglec-8 is a receptor within the family of I-type lectins and is well-studied in various immune cell-mediated diseases. Its role in cancer is poorly investigated, and no data of its impact on BC exist so far. We studied its expression levels in BC and found significant correlations with estrogen receptor status, grading and the prognostic factors Gal-7 [37] and TA-MUC1 [36]. The prognostic relevance of Gal-7 and TA-MUC1 was influenced by the expression levels of Siglec-8. Furthermore, Siglec-8 knockdown led to a reduced Gal-7 expression in cell culture experiments. Literature research reveals a possible role of Siglec-8 in the extra- to intra-cellular Gal-7-shuttling. Due to Siglec-8’s sialic acid binding capacity, it might also interact with TA-MUC1.

Furthermore, in in vitro experiments in BC cell lines, Siglec-8 expression could be upregulated by a PPARγ-agonist, suggesting a role of PPARγ in Siglec-8 regulation. Further investigation on interactions and regulation are needed to evaluate a possible role of Siglec-8 as a therapeutic target and predictive factor in BC.

## Figures and Tables

**Figure 1 ijms-22-02000-f001:**
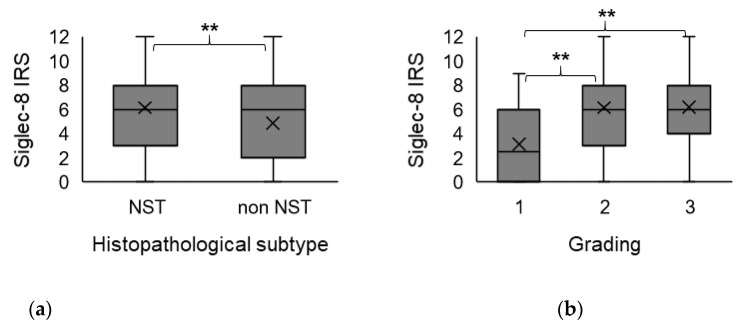
Association of Siglec-8 expression with histopathological subtype and tumor grading. Boxplots of the mean Siglec-8 IRS ± standard deviation (SD) dependent on histopathological subtype (**a**) and tumor grading (**b**) are shown. (**a**) In non-NST tumors, Siglec-8 expression is significantly lower than in NST tumors. (**b**) Tumors with G2/3 grading show a significantly higher Siglec-8 expression compared to G1 tumors. NST = no special type. IRS = immune reactivity score. SD = standard deviation. G = grading. ** indicates a *p*-value < 0.01.

**Figure 2 ijms-22-02000-f002:**
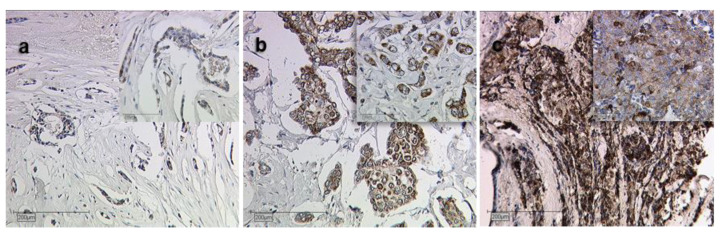
Siglec-8 expression dependent on tumor grading. Exemplary immunohistochemical staining results of Siglec-8 in grade 1 (**a**), 2 (**b**) and 3 (**c**) breast cancer are shown. Magnification: main images ×10, image sections ×25.

**Figure 3 ijms-22-02000-f003:**
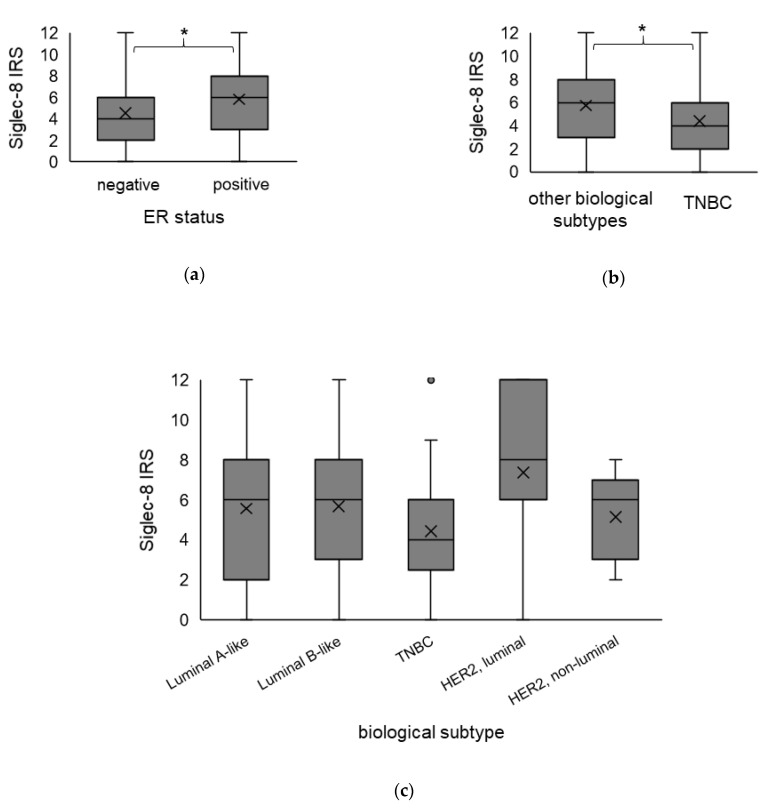
Association of Siglec-8 expression with the biological subtype. Boxplots of the mean IRS of Siglec-8 staining ± SD dependent on ER status (**a**), in TNBC compared to all other biological subtypes (**b**) and in the specific biological subtypes in detail (**c**), are shown. ER-positive tumors show a higher Siglec-8 expression than ER-negative tumors. Siglec-8 expression in TNBC is significantly lower than in the other biological subtypes. IRS = immune reactivity score. ER = estrogen receptor. TNBC = triple-negative breast cancer. HER2 = human epidermal growth factor receptor 2. SD = standard deviation. * indicates a *p*-value < 0.05. The dot in the Figure 3c indicates an outlier value.

**Figure 4 ijms-22-02000-f004:**
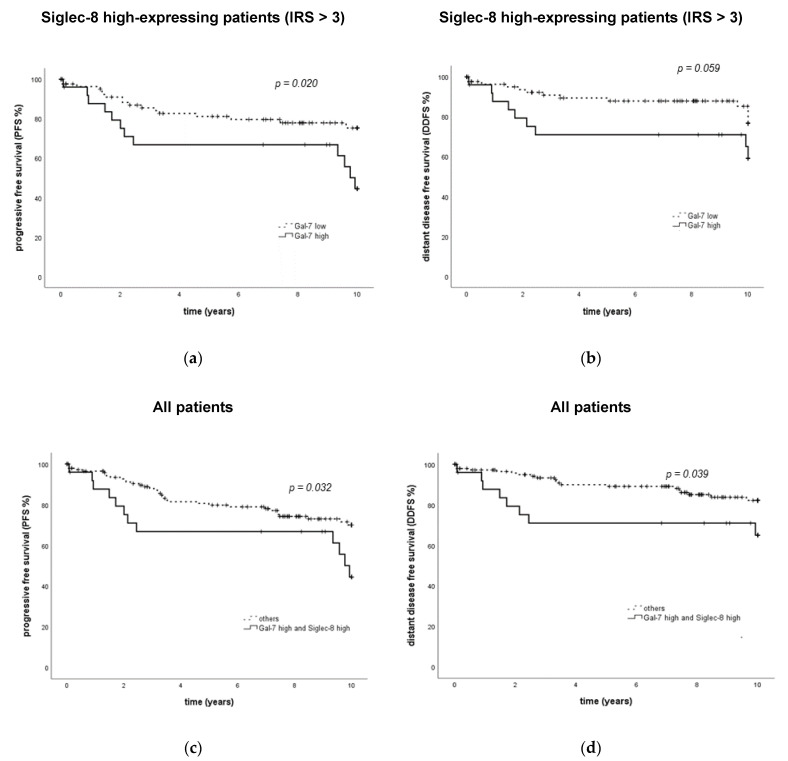
PFS and DDFS in subgroups defined by the combination of Gal-7 and Siglec-8 expression. Kaplan–Meier analyses of PFS (**a**,**c**) and DDFS (**b**,**d**) in subgroups defined by Gal-7 and Siglec-8 expression are shown. In Siglec-8-positive patients, PFS and DDFS differ significantly regarding Gal-7 expression (**a**,**b**), whereas Gal-7 was not significantly associated with PFS and DDFS in Siglec-8-negative patients. Patients in the “Gal-7 high/Siglec-8 high” showed a significantly impaired PFS (**c**) and DDFS (**d**) compared to all other subgroups. PFS = progression-free survival. DDFS = distant disease-free survival. Gal = galectin. IRS = immune reactivity score.

**Figure 5 ijms-22-02000-f005:**
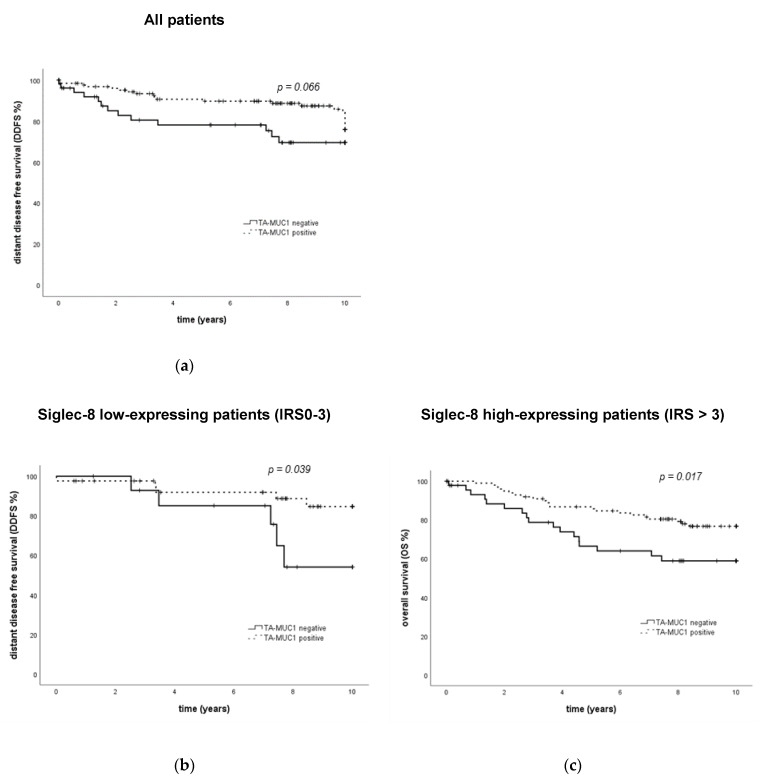
Clinical outcome of breast cancer patients regarding TA-MUC1 expression and in subgroups defined by the combination of TA-MUC1 and Siglec-8 expression. Kaplan–Meier analysis of DDFS shows an association of borderline significance between DDFS and TA-MUC1 expression in the overall cohort (**a**). In patients with low Siglec-8 expression, TA-MUC1 positivity is associated with an improved DDFS (**b**). However, in patients with high Siglec-8 expression, TA-MUC1 positivity is associated with an improved OS (**c**). TA-MUC1 = tumor-associated mucin-1. DDFS = distant disease-free survival. OS = overall survival. IRS = immune reactivity score.

**Figure 6 ijms-22-02000-f006:**
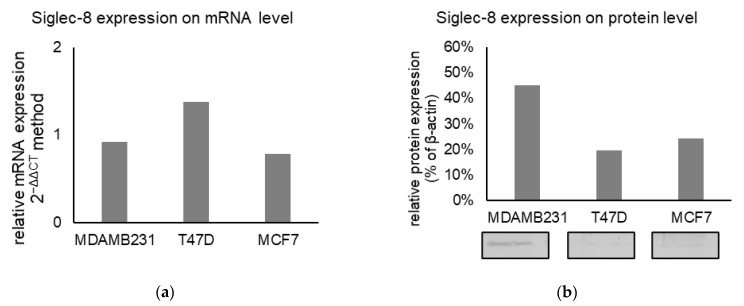
(**a**) Mean relative Siglec-8 expression on mRNA level (calculated with 2^−ΔΔCT^ method) in different cell lines, Glyceraldehyde-3-Phosphate Dehydrogenase (GAPDH) was used as endogenous control for ΔCT-values and the results are means of triplicates. (**b**) Mean relative Siglec-8 expression on protein level (as % of βActin expression) in different lines. Beta-Actin was used as a loading control and for normalization and results are the mean of triplicates. Exemplary Western Blots are displayed below the graph.

**Figure 7 ijms-22-02000-f007:**
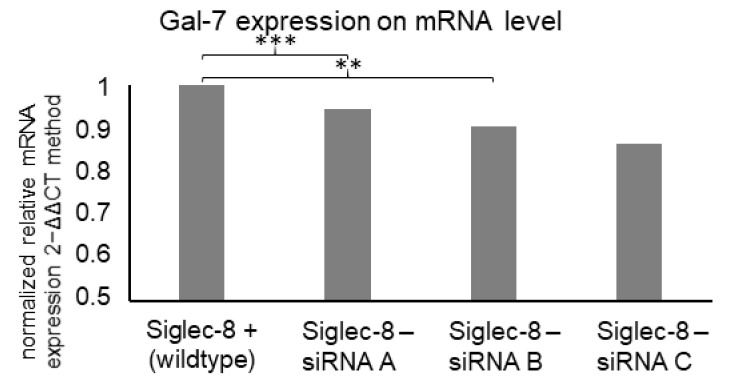
Mean relative Gal-7 expression on mRNA level (calculated with the 2^−ΔΔCT^ method) in MCF7 cells dependent on Siglec-8 silencing. Expression in control cells is displayed and expression in cells, where Siglec-8 was silenced with 5 nM siRNA A, B and C for 48 h, is normalized on the expression in control cells. GAPDH was used as endogenous control for ΔCT-values and the results are means of triplicates. siRNA A and siRNA B led to significantly reduced Gal-7 expression. ** *p*-value < 0.01, *** *p*-value < 0.001.

**Figure 8 ijms-22-02000-f008:**
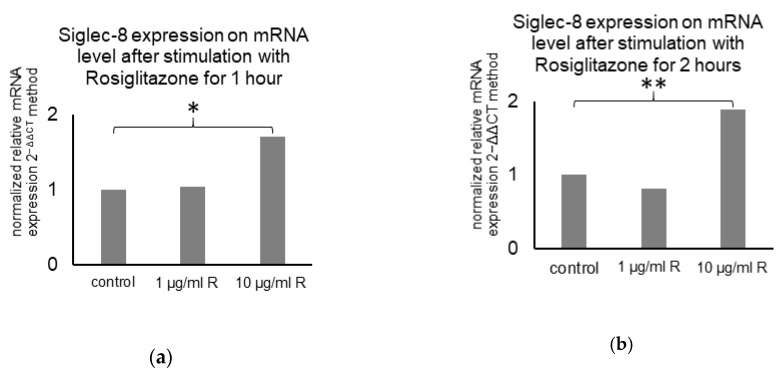
Mean Siglec-8 expression on mRNA level (calculated with the 2^−ΔΔCT^ method) in MCF7 cells dependent on PPARγ-stimulation. Expression in control cells is displayed and expression in cells stimulated with 1 or 10 µg/mL Rosiglitazone for 1 (**a**) and 2 (**b**) hours is normalized on the expression in control cells. GAPDH was used as endogenous control for ΔCT-values and the results are means of triplicates. * indicates a *p*-value < 0.05 and ** < 0.01.

**Table 1 ijms-22-02000-t001:** Staining results of Sialic Acid Binding Ig Like Lectin 8 (Siglec-8). IRS = immune reactivity score. NA = not applicable, staining could not be evaluated due to technical issues.

Siglec-8 Expression		
IRS	*n*	% of All
0	11	4.7
1	11	4.7
2	32	13.6
3	20	8.5
4	23	9.8
6	50	21.3
8	45	19.1
9	12	5.1
12	22	9.4
NA	9	3.8
total	235	100

**Table 2 ijms-22-02000-t002:** Patients’ characteristics.

	Median	SD
**Age**	58.2	13.3
	***N***	**%**
**Histological subtype**		
NST	126	53.6
Non-NST	96	40.9
**Biological subtype**		
Luminal A-like	103	43.8
Luminal B-like	73	31.1
HER2-positive, luminal	17	7.2
HER2-positive, non-luminal	7	3.0
TNBC	31	13.2
NA	4	1.7
**Grading**		
Grade 1	17	7.2
Grade 2	90	38.3
Grade 3	55	23.4
NA	73	31.1
**Lymph node status (pN)**		
pN0	128	54.5
pN1	87	37.0
pN2	10	4.3
NA	10	4.3
**Tumor size (pT)**		
pT1 (≤2 cm)	160	68.1
pT2 (2–5 cm)	68	28.9
pT3 (>5 cm)	1	0.4
pT4 (with infiltration in the epidermis or the thoracic wall)	5	2.1
NA	1	0.4
**HER2 amplification**		
positive	24	10.2
negative	208	88.5
NA	3	1.3
**ER status**		
positive	192	81.7
negative	43	18.3
**PR status**		
positive	141	60.0
negative	94	40.0

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
