# Peer review of "First Evidence for a Role of Siglec-8 in Breast Cancer"

_ijms, 2021, doi:10.3390/ijms22042000_

Round 1
Reviewer 1 Report
The research looks at the possible role of Siglec-8 in breast cancer. This is a very interesting topic as so far many functions of Siglec-8 have been described, but in immunology and other cancers (renal and gastric). It has been suspected for years that Siglec-8 may play a role in breast cancer, particularly TNBC, but there has been no clear evidence of this.
I confirm that the text is original and that no evidence of a role for the Siglec 8 in breast cancer has been published so far.
The manuscript is very interesting. It is possible that the text is quite complicated and I actually agree that it may require minor corrections, but I am not able to specifically indicate what should be improved in it to make it easier to understand (maybe the topic is just difficult).
The conclusions clearly show that the degree of Siglec-8 expression is important as a prognostic factor in breast cancer, and perhaps in the future it will also prove to be a predictive element (e.g. for immunotherapy) or as an independent target for potential therapies.
Author Response
Response letter to Reviewer 1:
The research looks at the possible role of Siglec-8 in breast cancer. This is a very interesting topic as so far many functions of Siglec-8 have been described, but in immunology and other cancers (renal and gastric). It has been suspected for years that Siglec-8 may play a role in breast cancer, particularly TNBC, but there has been no clear evidence of this.
I confirm that the text is original and that no evidence of a role for the Siglec 8 in breast cancer has been published so far.
The manuscript is very interesting. It is possible that the text is quite complicated and I actually agree that it may require minor corrections, but I am not able to specifically indicate what should be improved in it to make it easier to understand (maybe the topic is just difficult).
The conclusions clearly show that the degree of Siglec-8 expression is important as a prognostic factor in breast cancer, and perhaps in the future it will also prove to be a predictive element (e.g. for immunotherapy) or as an independent target for potential therapies.
We thank you for this comment and your interest in this topic. Indeed, there are many information given in this overviewing paper and that can be difficult to understand all of our conclusions. We tried to improve this issue with adding a comprehensive conclusion in the article (new paragraph 5 “conclusion”). We hope that the central leaches of our studies will now emerge more clearly.

Reviewer 2 Report
The review named „First evidence for a role of Siglec-8 in Breast Cancer” refers to novel results of an up-to-date topic.
The manuscript is designed and written well. Only few little comments to the figures. Figure 7 describes the mRNA level for Gal-7. The significance parentheses above bar Siglec-8 + (wildtupe) in the graph begins not directly above the bar, but next to. Please, correct it. In the figure 8, almost the whole graph a) is overlaid by the other graph b). Please, fix it.
Author Response
The review named "First evidence for a role of Siglec-8 in Breast Cancer" refers to novel results of an up-to-date topic.
The manuscript is designed and written well. Only few little comments to the figures. Figure 7 describes the mRNA level for Gal-7. The significance parentheses above bar Siglec-8 + (wildtupe) in the graph begins not directly above the bar, but next to. Please, correct it. In the figure 8, almost the whole graph a) is overlaid by the other graph b). Please, fix it.
We thank you for your suggestions for improvement of our figures. We corrected the significance parentheses above bar Siglec-8+ wildtype as suggested in figure 7. Regarding figure 8, we positioned the graphs correctly in the uploaded manuscript.
